# “Blood for Blood”? Personal Motives and Deterrents for Blood Donation in the German Population

**DOI:** 10.3390/ijerph18084238

**Published:** 2021-04-16

**Authors:** Klara Greffin, Silke Schmidt, Linda Schönborn, Holger Muehlan

**Affiliations:** 1Department of Psychology, University of Greifswald, 17489 Greifswald, Germany; silke.schmidt@uni-greifswald.de (S.S.); muehlan@uni-greifswald.de (H.M.); 2Institute for Immunology & Transfusion Medicine, University Medicine Greifswald, 17489 Greifswald, Germany; linda.schoenborn@med.uni-greifswald.de

**Keywords:** blood donation, behaviour intention, motives, deterrents, representative sample

## Abstract

It is crucial to provide updated knowledge about blood (non-)donors, as it is necessary to design targeted interventions with the aim of retaining blood donors and thus contributing to a functioning health system. This study investigates the prevalence and socio-demographic patterning of lifetime blood donation, assessing blood donation intention within the next 12 months and exploring personal motives and deterrents of blood donation qualitatively in the German population. A face-to-face cross-sectional survey with 2531 respondents was conducted, representative of the German population in terms of age, gender, and residency. Closed as well as open questions were asked. Qualitative content analysis was used for coding the qualitative material. Basic descriptive statistics were conducted to address our research questions. More than one-third of the participants reported that they have donated blood at least once in their lifetime. Motives and deterrents were assigned to 10 domains with 50 main categories and 65 sub-categories. The most frequently stated motives for blood donation were “altruism”, “social responsibility”, and “charity”, whereas the most frequently stated deterrents were “health status”, “age”, and “lack of time”. This study provides information to tailor recruitment and reactivation strategies to address donors at different career steps—from non-donor to loyal donor.

## 1. Introduction

The limited availability of blood is a major concern for many countries around the world. To ensure adequate health care, a balance between voluntary blood donations and the need for blood in a population is crucial [1]. However, the number of people who donate blood decreases [2,3]. With the aging “baby boomer” generation, the demand for blood will likely increase in the next 10–15 years, when this population group shifts from being potential donors to those requiring most of the transfusions [3,4]. However, in the past ten years, transfusion demand decreased in many European countries, too, mostly due to an active reduction within patient blood management programs [5] or improved treatment options with a smaller need for blood. It is unclear whether this trend will continue, as some hospitals already implemented very restrictive transfusion triggers [4]. As further reductions may be limited, increasing the number of blood donations remains crucial. The strategy of building a reliable donor pool further offers valuable advantages: blood donation appointments can be allocated according to blood demand, which in turn allows a forecast to be made and the number of staff to be adjusted [6]. In addition, repeated donors are less likely to transmit infections, are motivated to recruit new blood donors, and have a positive impact on cost-effectiveness [6].

The imbalance between demand and supply also exists in Germany, where only 2–3% of the population donates blood [7]. This is aggravated by difficulties in finding adequate strategies to recruit or reactivate donors [8,9,10,11], as well as difficulties with the general donation process such as communication and staff skills [6,12]. In order to ensure blood supply, it is essential to develop a better understanding of motivators and deterrents in blood donation behaviour and how they affect different donor groups.

Since the 1950s [13,14], researchers have been interested in what motivates or prevents people from donating blood. Ever since, the research field has evolved, and until today there have been quite a number of studies that examine donor motivation and deterrents [15,16,17,18,19,20,21,22,23,24,25,26,27]. Although these studies utilize similar concepts, they are difficult to compare, because nomenclatures and the different donor groups are inconsistent [14]. In a systematic review, Bednall and Bove (2011) compared motivators and deterrents of donation with regard to donor career stages. They found that prosocial motivation, personal values, and convenience were the most frequently mentioned motivators for first-time and repeated donors, whereas low self-efficacy to donate, low involvement, perceived inconvenience, and lack of marketing communication were the most often stated deterrents in donors and non-donors. In addition, there is evidence that receiving financial remuneration does not motivate donors in the long run, while this is unresolved for items like health screening, vouchers, or tickets [9,17,20,28,29,30].

Blood donation behaviour does not only depend on deterrents and motivators but also on cultural, economic, and demographic factors, which is why it is essential to consider each country and region separately. There are several countries in which representative surveys have been conducted regarding the reasons why people donate blood or not [22,25,31].

Two representative studies exist that have explored motivators and deterrents of blood donation behaviour in Germany. In a survey from 1998, Riedel and colleagues examined attitudes towards blood donation in Germany in a sample of 2032 participants. The main findings were that 8% of participants donate regularly, 34% are willing to donate but do not, and 14% strictly refuse to donate. Men donate more regularly than women do, while women mention health reasons twice as often as obstacles to donating blood. However, there are no gender-specific differences in the general refusal of blood donation. Further, younger study participants and persons with higher education are more willing to donate blood. Thus, as of over 20 years ago, the typical blood donor in Germany was male, 39 years old or younger, and had a high school education [32].

In 2018, the German Federal Centre for Health Education published a representative study with a sample of 3836 participants that were surveyed using computer-assisted telephone interviewing. The main findings were that 47% of those surveyed had already donated blood once in their life. Surprisingly, 23% of the participants had donated within the past 12 months. What stood out was that only 29% of respondents in the 18–25 age group had ever donated blood. At the same time, this age group made up the largest active group, with 56% donating blood within the past 12 months. The main obstacles for donating blood were (a) health reasons/medication and (b) lack of time/have not thought about it [24].

The studies on German blood donation behaviour [24,32] are valuable but limited with regard to (a) a specification on a limited amount of constructs or (b) up-to-dateness, as well as (c) the use of quantitative methods and closed questions. Therefore, the aim of the present study was to (i) investigate the prevalence and socio-demographic patterning of lifetime blood donation, (ii) assess blood donation intention within the next 12 months, and (iii) explore personal motives and deterrents of blood donation in the German population with regard to different donor career types using a representative sample in an exploratory design.

## 2. Materials and Methods

### 2.1. Study Population and Recruitment Procedure

Questions assessing blood donation behaviour were part of a survey conducted by USUMA, a German enterprise for market and social research. The sampling procedure was chosen in order to collect a representative sample of the German population in terms of age, gender, and residency according to the federal state distribution. The final 2531 valid interviews were conducted by 223 interviewers, constituting approximately 11 interviews per interviewer.

To ensure representativeness, our sampling procedure was as follows: Initially, the Federal Republic of Germany was electronically divided according to intra-municipal territorial sections, ending up with 53,000 defined regional areas, each including 700 households on average. In a next step, these areas were stratified by regional districts and geospatial types. From the resulting strata, target households were selected, applying the random-route method. Thus, interviewers received a street address as an initial sample point. From there, every third household was detected until 20 valid private addresses per each sample point had been identified according to a given route inspection plan.

This procedure resulted in 5093 valid addresses. From each household, one participant aged 14 years or older was randomly chosen and contacted up to three times. In sum, 731 households and 134 subjects had to be excluded because they were not at home. Moreover, 840 households and 804 subjects refused participation, 25 subjects were out of town, 22 persons were sick, and six interviews were not valid. Finally, 2531 fully valid interviews remain for our analyses.

Information regarding blood donation was assessed within multi-topic face-to-face interviews. Further questions, that are not related to blood donation behaviour were implemented by other research groups and are not part of the current study.
(a)*Socio-demographic factors:* Age, sex, education, employment, (household) income, religious confession, nationality, as well as family and partnership status were assessed by a socio-demographic questionnaire.(b)*Blood donation behaviour and intentions:* Respondents were asked to decide whether they (1) did not, so far, imagine donating blood, (2) could imagine donating blood but did not manage to do so, (3) had already tried to donate blood but were deferred due to ineligibility, or (4) had already donated blood at least once in their lifetime (participants indicated which of these statements they agree with). Respondents who stated that they already had donated blood were referred to as “donors” (4), and the remaining respondents were referred to as “non-donors” (1 and 2) or deferred donors (3). The intention to donate blood within the next 12 months was assessed on a scale from 1 = “definitely not” to 6 = “definitely yes” (see Table A1 in Appendix A). We assigned the respondents according to their respective statements about future donation intentions to two different groups: respondents who intended to donate blood “rather”, “probably”, or “definitely yes” were referred to as “intenders”, respondents who intended to donate blood “rather not”, “probably not”, or “definitely not” were referred to as “non-intenders”.(c)*The personal reasons (motives and deterrents) for donating blood were investigated with two open questions.* The first question was: “Based on your answer, we would like you to describe in your own words, what, so far, your personal reasons were for donating or not donating blood in the past. Please try to answer this question as exactly as possible.” The second question was: “Based on your answer, we would like you to describe in your own words, what your personal reasons are for donating blood or not in the future. Please try to answer this question as exactly as possible.” Each open question was posed subsequently to the corresponding closed question to learn more about personal motives for (a) past (non-)donation behaviour or (b) future (non-)donation intention, respectively (see Table A1).

### 2.2. Coding Procedure

Qualitative material of the personal motives and deterrents for donating blood in the past or in the near future was analysed using qualitative content analysis (Mayring, 2014, 2015). To analyse the open question concerning personal reasons (motives and deterrents) for or against donating blood, initial categories were separately derived inductively by the two main researchers (KG and HM) and discussed until consensus had been achieved. After the motives for the first 1000 respondents were coded, the codes and discrepancies were discussed, and the category system was adapted accordingly to reach consensus between both raters. The resulting category system was then used by two other coders (GD and SH) to rate the personal reasons for all 2531 respondents. It turned out that the resulting category system comprised too many domains. Moreover, those identified lack both, comparability with previous research and the potential for delineating practical implications. Thus, we decided to conduct a second round of the coding process, applying an adopted approach to account for the aforementioned shortcomings. Therefore, as a starting point, a set of categories was adopted from the meta-analysis provided by Bednall and Bove (2011). Subsequently, we repeated the coding process with the first 1000 respondents and additional categories were derived inductively once again by the main researchers separately. Afterward, we checked agreements and discrepancies of the coding until full consensus was reached. Finally, this category system was applied by two new coders (AM and AT) to rate the personal reasons for all 2531 respondents once again. The percentage of agreement was calculated (87.4%). Finally, each discrepancy was discussed by KG and HM until consensus was reached for each code, respectively. The final category system was applied to categorize both answers to the open question about past (non-)donation behaviour and future (non-)donation intentions.

### 2.3. Statistical Analyses

We used a multi-level sample design, as this ensured that every household in which target persons lived was equally likely to be included in the sample. If the household was selected, persons in larger households had a lower probability of selection than persons in small households. This effect was balanced by a design weighting. Nonresponse also led to a distortion of the distribution of various characteristics compared to the population. This is reduced by an adjustment weighting. Descriptive statistics were used to describe the respondents’ socio-demographic characteristics as well as past and intended blood donation behaviour. Effect size measures for contingency tables (Phi, Cramer’s V) were used to provide standardized indicators of group differences in motives and deterrents between different types of blood donors [33]. All quantitative statistical analyses were conducted using IBM SPSS Statistics 25.0 [34].

### 2.4. Ethical Approval

The survey has been performed in accordance with the ethical standards as laid down in the 1964 Declaration of Helsinki declaration and its later amendments or comparable ethical standards. Informed consent was obtained from all individual participants included in the study. The ethical approval for this study was given by the ethics committee of the Medical Faculty of the University of Leipzig in Germany (file number: 418/17-ek).

## 3. Results

### 3.1. Respondents

The total sample consisted of 2531 respondents and was representative of the German population in terms of age, gender, and residency. The participants’ mean age was 48.6 years (SD = 18.0), with ages ranging from 14 to 93 years. Slightly more than half of the participants were female (51.0%). Approximately one-third of the participants had no or the lowest formal qualification (32.4%), 38.6% had an intermediary secondary qualification, 15.9% had a higher secondary qualification, and 8.8% had a university degree. Approximately 19.6% of the participants reported having a household income of up to €1500 a month, nearly one-third of the participants reported having a household income of up to €2500, and 23.7% have a household income of up to €3500 or more than €3500 a month. The majority of the participants reported belonging to the Christian confession (68.2%) with 37.1% being Protestant and 31.1% being Catholic. Table 1 displays the socio-demographic characteristics of the total sample. 

### 3.2. Prevalence of Blood Donation

Nearly two-thirds of the respondents stated that so far they had never donated blood (“Non-Donors”: *n* = 1526, 65.6%)—either they could not imagine donating blood (*n* = 1032, 40.8%), or they could imagine it but had not managed to donate blood yet (*n* = 494, 19.5%). Additionally, 5.1% (*n* = 130) had already tried to donate blood but were not eligible to do so. The remaining 34.4% of the participants (*n* = 870) stated that they had already donated blood at least once (“Donors”).

Similarly, more than two-thirds of the respondents indicated that they do not intend to donate blood within the next 12 months (“Non-Intender”: *n* = 1752, 69.2%), with the majority answering with “definitely not” (*n* = 1035, 40.9%). Accordingly, less than 30% of the participants (*n* = 708, 28.0%) reported that they intended to donate blood within the next 12 months, equally distributed into “rather likely” (*n* = 245, 9.7%), “probably likely” (*n* = 212, 8.4%), and “definitely yes” (*n* = 252, 10.0%; see Figure 1 and Figure 2 as well as Table A2).

Table 2a,b provides an overview of the socio-demographic characteristics of past non-donors, deferred donors or donors (Table 2a), and non-intenders or intenders (Table 2b), respectively.

### 3.3. Personal Motives and Deterrents for Blood Donation

#### 3.3.1. Category Framework

The category framework consists of ten domains containing three to eleven categories, respectively, and one additional “miscellaneous” section with three categories:(1)“Ineligibility”: This domain contains all statements relating to reasons for the perceived inadequate suitability of blood donation. Statements were assigned to two categories, distinguishing between specific and unspecific reasons: (a) “specific reasons” include subcategories such as “age”, “health”, “pregnancy”, and “other” (e.g., “trips abroad”), whereas (b) “unspecific reasons” encompassed statements concerning a lack of “eligibility” without giving any specific reasons (e.g., “I am not eligible”).(2)“Impact and Effect”: This domain comprises all statements concerning the anticipated impact or effect of donating blood and contains three categories: (a) “physical consequences” encompasses expected positive physical effects (e.g., “I’m doing something good for myself, my body”), as well as negative physical effects (e.g., “my cardiovascular system can’t handle that”), and health risks; (b) “mental well-being” comprises both positive as well as negative psychological effects for the donor (e.g., “it’s (not) good for me”).(3)“Fear and Aversion”: This domain included two main categories. The first category, (a) “fear”, is again split into “fears in general”, i.e., without further specification (e.g., “I am afraid”) and four categories for specific fears, existing or anticipated “fear of needles”, “fear of blood”, “fear of pain”, and “fear to donate” (e.g., “I’m afraid of blood getting taken”). Any statements relating to specific other fears are assigned to the category “other fears” (e.g., “fear of doctors”). The second category is (b) “aversion”, which is subdivided into three subcategories concerning a personal aversion to “needles”, “blood”, or “other things”.(4)“Obstacles and Barriers”: This domain comprises all statements relating to possible logistical or organizational obstacles, assigned to the categories (a) “lack of information and knowledge” (e.g., “I wouldn’t even know where and how I could donate”), (b) “lack of possibilities or opportunities” (e.g., “no opportunity nearby”), (c) “organization/effort” involved (e.g., “too cumbersome, complicated”), and (d) “time/lack of time” (e.g., “I hardly have time for that”), or due to (e) “personal reasons”(e.g., “I can’t set it up”).(5)“Norms”: This domain consists of three core categories. First, (a) “reciprocity” is subdivided into four sub-categories. “General reciprocity” refers to statements concerning the recognition of the norm of reciprocity, because of past or possible future health care use (e.g., “blood for blood”). “Future-orientated reciprocity” is directed to the expectation of increasing the future possibility to receive someone else’s blood by donating blood (e.g., “a situation could come up in which one needs blood”). “Past-orientated (self) reciprocity” includes statements recognizing the norm of reciprocity because of having received someone else’s blood in the past (e.g., “to give back something I received”), whereas “past-orientated (friends and family) reciprocity” includes statements recognizing the norm of reciprocity because friends or family have received someone else’s blood (e.g., “in return for the blood my child received”). The second norm-based category, (b) “altruism”, consists of statements of the unconditional necessity of helping people (e.g., “that is the way in which I can save lives”). Third, (c) “feelings of obligation, social conscientiousness, and responsibility” is the general subjectively felt obligation and social norm or expectation to donate blood (e.g., “to do something for the community”). Other categories are (d) “religious beliefs”, including religious or denominational reasons, and (e) “personal beliefs”, including personal beliefs that are not covered by other normative beliefs (e.g., “it is important to me personally”). The category “important/necessary/meaningful/good cause” is concerned with a generally described relevance to donate blood “because it is important” because of a “vocational affiliation” (e.g., “through my work in the clinic”) or “surrendering responsibility”, addressing statements of existing insight of why individuals should become active (e.g., “enough others do it”), or because the personal need of one’s blood (e.g., “I need my blood myself”).(6)“Image and Experience”: This domain contains all statements that relate to specific images of or experiences with characteristics of blood donation. Respective categories are to “compensate for the lack of blood products/support the health care system” (e.g., “demand is constantly rising”), “lack of trust” due to media reports of fraud and profiteering (e.g., “because I don’t want to support such profiteering”), “no need present” (e.g., “There is no lack of blood”), or “rare blood type”—representing the knowledge of the rarity and thus the relevance of the donation (e.g., “important, I have a rare blood type”). Moreover, other categories are “advertisement campaign/phone call/appeal” (e.g., “promo day”), “(missing) previous experience or habit” (e.g., “I am a permanent donor and regularly donate blood”), “curiosity” (e.g., “I was curious”), or “absence of disadvantages” (e.g., “it doesn’t hurt me”).(7)“Benefits and Incentives”: This domain covers all motives mentioning compensatory measures. Respective types of motives cover a wide range of benefits and incentives; categories include (a) “blood donor card”, (b) “financial compensation”, (c) “determination of the blood type”, (d) “health check/screening” (free of charge), (e) “exemption from work/school”, as well as (f) “other services” including any other services or discount the donor receives (e.g., “extra holiday”).(8)“Conditions”: This domain comprises all statements of specific conditions in which blood donations would be given. This includes categories such as (a) “for personal need only” (e.g., “for myself alone”), (b) “only for family/important others/if person is known” (e.g., “why should I? if, then only for relatives”), or (c) “in case of a disaster/emergency/personal experience”, due to special demand or circumstances (e.g., “train accident”).(9)“Psychological aspects”: This domain comprises all statements related to attitudes, volition, and behaviour. It contains categories such as (a) “will be made up for/is planned”; (b) “not ready yet”; (c) “no interest/no will”; (d) “indifference/passivity/comfort”; (e) “not thought about it”; (f) “social aspect/peer group movement/personal influence and advice”, which contains statements of social motivation (e.g., “my ex-girlfriend took me there”); and (g) “refusal to donate blood”, addressing explanations due to the belief that blood donation is meaningless or not important.(10)“Missing points of contact”: This domain comprises all statements relating to missing triggers or contact points. It includes the categories (a) “no request/appellation/call”, which concerns the fact that the respondent has not explicitly been asked, called upon to donate, or addressed personally (e.g., “someone should ask me about it once”). Another reason may be a missing special occasion or reason to donate blood, encoded in the category (b) “missing reason/occasion” (e.g., “there hasn’t been an occasion to do it yet”). Finally, (c) “for no reason”, covers any statement where the respondents present a lack of awareness of their own motivations and obstacles.

(+) “Miscellaneous”: A final set of categories, not to be referred to as a “domain”, containing all remaining statements. (a) “Other” for all statements that cannot be assigned to the other categories, (b) “don’t know”, if the respondent does not know or cannot remember anymore, and (c) “not specified”, if a statement is missing or was actively refused.

#### 3.3.2. Frequencies of Reasons

Overall, 5681 personal motives and deterrents of blood donation in the past and the near future were stated by 2531 participants. Reasons were assigned to 11 domains with 50 main categories and 65 sub-categories (see Table 3). Frequencies for all blood donation motivators and deterrents in the total sample are depicted in Table 4a. The number of statements within a single category equals the number of cases, but this is not the case for the cumulative number of statements within a domain, as one reason could be assigned multiple codes if deemed necessary.

The most frequently stated motives for blood donation in the past (across a lifetime) are all assigned to the domain “norms”, covering 812 statements in total. Amongst these, the most frequently reported motive was “altruism” (*n* = 439, 17.3%), an unconditional necessity of helping people and saving lives. For example, participants indicated that they see blood donation as a way to save lives. Another frequently stated motive was “social responsibility” (*n* = 113, 4.5%), generally described as a subjective obligation of the social norm or expectation to donate blood. The third most often mentioned reason was “charity” (*n* = 94, 3.7%), emphasizing the relevance of donating blood in somewhat general terms (e.g., because it’s “important”, “necessary”, “meaningful”, or a “good cause”)—followed by “compensate for lack of blood products” on rank 4 (*n* = 69, 2.7%) and incentive in terms of a financial “compensation” on rank 5 (*n* = 66, 2.6%). The frequencies of statements of any other motive drop below 2.5%.

With respect to the most frequently stated motives for intentions to donate blood in the near future (12 months), the first four categories appear to be the same as for previous blood donation behaviour, with a slightly different rank ordering, that is “altruism” (*n* = 330, 13.0%), “charity” (*n* = 81, 3.2%), “social responsibility” (*n* = 77, 3.0%), and “compensate for lack of blood products” (*n* = 70, 2.8%), followed by “future-orientated reciprocity” on rank 5 (*n* = 60, 2.4%). Detailed information is provided in Table 4b.

Compensation for past donations was more important to men than to women (1.9% vs. 0.7%). With regard to future donation intention, women state altruistic motives more often than men (7.2% vs. 5.9%) and are more often willing to donate only for significant others (12.0% vs. 6.0%; see Table A3, Table A4 and Table A5).

#### 3.3.3. Frequencies of Deterrents

Primary deterrents for blood donation in the past (across a lifetime) are assigned to diverse domains, such as specific reasons of being ineligible for blood donation (e.g., age or health status; *n* = 468, 18.9%), any “psychological aspects” (e.g., indifference/passivity; *n* = 440, 17.4%), or several “obstacles/barriers” (e.g., no opportunity to donate blood; *n* = 338, 13.2%). On the level of single categories, the most frequently stated deterrent was “health status” (*n* = 302, 11.9%), followed by “not thought about it” (*n* = 161, 6.4%), “lack of time” (*n* = 143, 5.6%), “no interest/no will” (*n* = 127, 5.0%), and “age” (*n* = 122, 4.8%).

With respect to the most frequently stated reasons not to donate blood in the near future (12 months), most of the categories mentioned before are included once more among the highest ranked statements, but with a more clearly differing rank order: “health status” on rank 1 (*n* = 346, 13.7%), “age” on rank 2 (*n* = 292, 11.5%), and “lack of time” (*n* = 152, 6.0%) and “fear of needles” (*n* = 81, 3.2%) on ranks 3 and 4, respectively, and “no interest/no will” (*n* = 70, 2.8%) on rank 5 (see Table 4b).

“Health” (8.1% vs. 3.8% for past behaviour; 9.0% vs. 4.7% for future donation behaviour) and “fear of needles” (2.6% vs. 1.0% for past behaviour; 2.4% vs. 0.8% for future donation behaviour) were more often stated as deterrents by women than by men (see Table A3, Table A4 and Table A5).

## 4. Discussion

Based on a representative interview–survey of the German population, this study provides updated insights into personal motives and deterrents for past as well as future blood donation behaviour.

### 4.1. Main Results

As a first result, motives and deterrents were assigned to a category system consisting of 10 domains (and one additional “miscellaneous” section) with 50 main categories and 65 sub-categories. This category system was initially based on a previous meta-analysis by Bednall and Bove (2011) and was supplemented by inductively derived categories from our analysis by adding missing categories and further differentiating selected motivators and deterrents. An example is the category “lifestyle barriers”, which Bednall and Bove used. In our present study, we differentiated this category into “lack of information and knowledge”, “lack of possibilities or opportunities”, “organization/effort”, “time/lack of time”, and “personal reasons”. Two examples for additional categories not listed in the meta-analysis of Bednall and Bove (2011) are “physical consequences (positive and negative physical effects)” and “mental well-being (positive and negative psychological effects)”, which came as no surprise, since previous research has frequently reported the (expected) physical and mental impact of blood donation [35,36,37,38]. Notably, studies investigating the potential positive influence of blood donation on well-being are comparatively a more recent development [39,40,41]. The chosen approach in this study is a valuable step towards the harmonization of nomenclatures and hence better comparability of data.

### 4.2. Motives and Deterrents in Different (Non-)Donor Groups

Our data indicate that most of the donors and donor intenders are motivated by altruism, social responsibility, or a good cause. The most common deterrents stated by non-donors and donor non-intenders are “age”, “health and physical conditions”, “fear”, “organization”, and “passivity”, while deferred donors were mostly unable to donate due to health issues.

From these reasons, strategies can be derived that may increase willingness and eligibility to donate blood: First, educational materials should be easier accessible and understandable. Digital communication technologies (e.g., apps) may enlarge the accessibility, while the involvement of different donor and non-donor groups may be beneficial for the development of suitable educational material. The Blood Donation Fears Inventory [42] can be used to assess different types of fear felt by current and potential donors. After identifying the degree of fear, it could be addressed through educational material about potential misconceptions, tailored communication strategies, and “reality checks” (e.g., open-door days). By including strategies inspired by psychotherapeutic techniques, addressing fear in blood donation could also be a great example of the benefit of interdisciplinary cooperation. Organizational barriers such as lack of time and opportunities could be addressed by suitable opening hours for all donor groups (e.g., students may be more flexible to donate than working parents with children) and complementary offers such as free Wi-Fi or childcare. In addition, a short distance to the nearest donation centre is beneficial [43]. Finally, passivity describes statements like “I have not thought about it”, stated by almost 11% of non-donors. To gain their attention they need to be addressed through (personal) requests, appellations, or phone calls in order to heighten the probability that they will donate blood.

In our sample, monetary incentives were not a main motive to donate blood, which is consistent with the findings by Costa-Font et al. (2013, [44]). However, monetary incentives were significantly more often stated as a motive by previous (male) donors and donation intenders than as a barrier by non-donors or non-intenders. Non-monetary incentives, such as health checks, seem to have a minor importance with regard to previous or future donation motivation, which is in line with Goette et al. (2009, [45]).

### 4.3. Donation Behaviour in Different Groups

Taking a closer look at current donors, deferred donors, and non-donors of our sample, a high proportion of those surveyed do not intend to donate blood in the near future: 86.9% of deferred donors and 86.1% of current non-donors state that they will not donate blood within the next twelve months. In contrast, 61.0% of current donors indicate the intention to continue blood donation behaviour. Due to the anticipated non-donation behaviour, we suggest that it should be studied whether the self-reported reasons against a donation are real reasons for donor ineligibility. For example, the main reasons against a future donation were ineligibility due to health issues or age. It is of interest whether the perceived health status is actually a reason for not donating blood. With regard to age, donors, deferred donors, and non-donors show a similar distribution at each age group. Interestingly, age differences occur in the context of future donation behaviour: While donation intenders predominate in the age groups 18 < 25, 25 < 35, and 45 < 55, donor non-intenders represent the majority from the age group of 55+. For this reason, blood donation centres should provide easily accessible and understandable information about donation-related age restrictions. In general, deterrents for future donation intention should be further studied in order to identify potential gaps of knowledge that should be addressed by educational interventions.

The majority of deferred donors (61.3%) were female. This result is in line with previous international studies [46,47,48,49,50]. In the light of low future donation intention in the group of deferred donors, this result highlights the need for donation-related information that is designed in a gender-sensitive way to limit the long-term consequences of a deferral. We also learned from our data that women and men differ in terms of motivators and deterrents. In conclusion, recruitment and reactivation material should also be designed in a gender-sensitive way in order to address the gender-specific motives.

### 4.4. Comparison with Previous Studies

Compared to the study by Riedel et al. (2000), the distribution of our data is different with regard to previous blood donor behaviour. In the study from 2000, 38% of the respondents had already donated blood and 34% were generally willing to do so, whereas 29% of the respondents were not eligible to donate for health reasons or refused to donate. The proportion of those who cannot imagine donating blood is higher in our study (40.8%), while the proportion of those who would be willing in general has fallen considerably (19.5%). About the same number of respondents have already donated blood (34% vs. 38%). Deferred donors were only explicitly recorded in our study, whereby it is not clear in the study by Riedel and colleagues whether the participants who stated that they cannot donate for health reasons have already attempted to donate blood. Riedel et al. (2000) describe the typical blood donor as a man who belongs to the younger age group up to 39 years, tends to have a higher education, and lives in the western federal states. In the current study, we also find that more men than women donated blood successfully and more men than women intend to do so. With regard to age, we find that donors can be found across all age groups, but most frequently in the 45–54 age group for both previous behaviour and future intention. Interestingly, the shift in age group coincides with the time passed between the two studies. A higher level of education also supports blood donation behaviour in our sample. 

The BZgA study from 2018 records previous blood donation behaviour in a binary way: 47% of respondents state that they have already donated blood once, while 53% negate this question. The proportion of those who have already donated blood is thus significantly higher than in the study by Riedel and colleagues and our study. This circumstance may possibly be based on the larger sample size (*n* = 3836 in the BZgA study compared to 2081 in the study by Riedel et al. or *n* = 2531 in this study). All three studies show that, proportionally, more men have already donated blood than women have.

With regard to motives and barriers, our results partly match with the results from previous studies: Bednall and Boves (2011) describe that the convenience of the collection site, prosocial motivation (altruism and collectivism), and personal values were the most frequently stated motives for past behaviour, whereas low self-efficacy, low involvement, and inconvenience were the most common deterrents [14]. In another study on the German population by the Federal Centre for Health Education, health reasons (41%) and a lack of time (33%) were reported as the most stated deterrents to blood donation (in the past 12 months). Riedel et al. (2000) found that 15% of the participants could not donate because of health issues, which matches with the findings in our current study [32]. Nevertheless, none of these studies reported motives and deterrents for future donation intention.

### 4.5. Strengths and Limitations

Our study has certain strengths and limitations. A strength is that it was based on a representative sample in terms of sampling, recruitment, and characteristics. Furthermore, data were collected directly via face-to-face interviews using closed and open-ended questions. Consequently, we combined quantitative and qualitative data analysis to evaluate the data. In this context, we combined a deductive and inductive approach to adopt a pre-existing category system, which enabled us to compare our data with prior studies and provide new insights. The qualitative approach to assessing motives and barriers is more time-consuming than using quantitative scales. In the context of this explorative study, however, we had the opportunity to take into account all individual motives and barriers and to enable multiple answers. This procedure is particularly recommended if one is interested in the motives and barriers of previously neglected groups (e.g., deferred donors), which may differ from the information provided by donors and non-donors. The use of qualitative methods is therefore in line with our goal of improving the mapping of motives and barriers of different donor groups, as it makes it possible to capture the narratives of blood donation, which are underlying persistent societal changes. Thus, on the level of everyday subjective experiences, individual scripts, personal reasons, and self-explanations for (not) donating blood change as well. In addition, we wanted to provide a broader spectrum of information than the previous studies with the aim of being able to derive interventions more easily from this. Finally, we assessed a spectrum of reported personal reasons for and against a blood donation in the past as well as future, which is a decisive advantage over other studies.

There are some critical aspects that should be considered while interpreting our data: We posed an open question to gain information about the motives and deterrents of blood donation rather than conducting in-depth interviews, which is why our qualitative data lacks contextual information. All motives and deterrents are based on self-reported data provided in face-to-face interviews, so there is a risk that socially desirable answers were given. Finally, our study did not have a longitudinal design, so there is no follow-up linkage between the assessed donation intention and actual donation behaviour within the next twelve months.

## 5. Conclusions

Ensuring sufficient blood donations is expected to become more and more challenging given the demographic change e.g., in Germany and the growing need for blood supplies due to complex surgeries, as well as dwindling blood supplies shown by recent studies. Thus, based on a representative survey of the German population, this study provides information to tailor recruitment and reactivation strategies to address donors at different career steps—from non-donor to loyal donor—as it is important to reach out to everyone. In future studies, we aim to identify different donor career types to learn more about prior circumstances that lead to future intentions.

## Figures and Tables

**Figure 1 ijerph-18-04238-f001:**
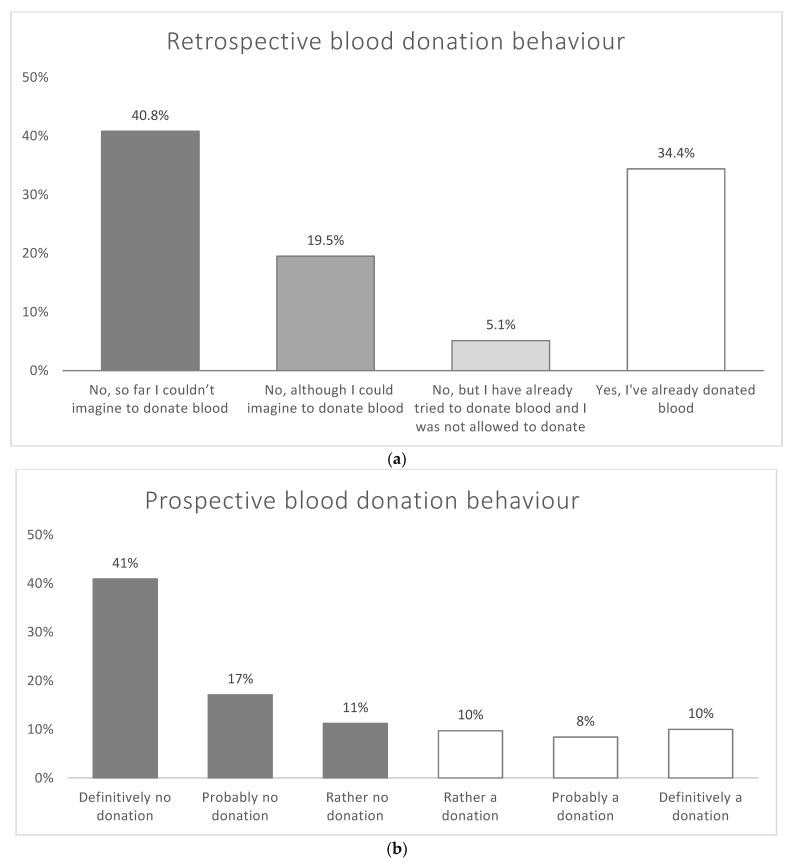
(**a**) Self-reported retrospective (lifetime) blood donation behaviour—results for total sample (*n* = 2531, weighted). Absolute frequencies and cumulative percentages vary as a function of the amount of missing data for each variable. (**b**) Self-reported prospective (12 months) blood donation behaviour—results for total sample (*n* = 2531, weighted). Absolute frequencies and cumulative percentages vary as a function of the amount of missing data for each variable.

**Figure 2 ijerph-18-04238-f002:**
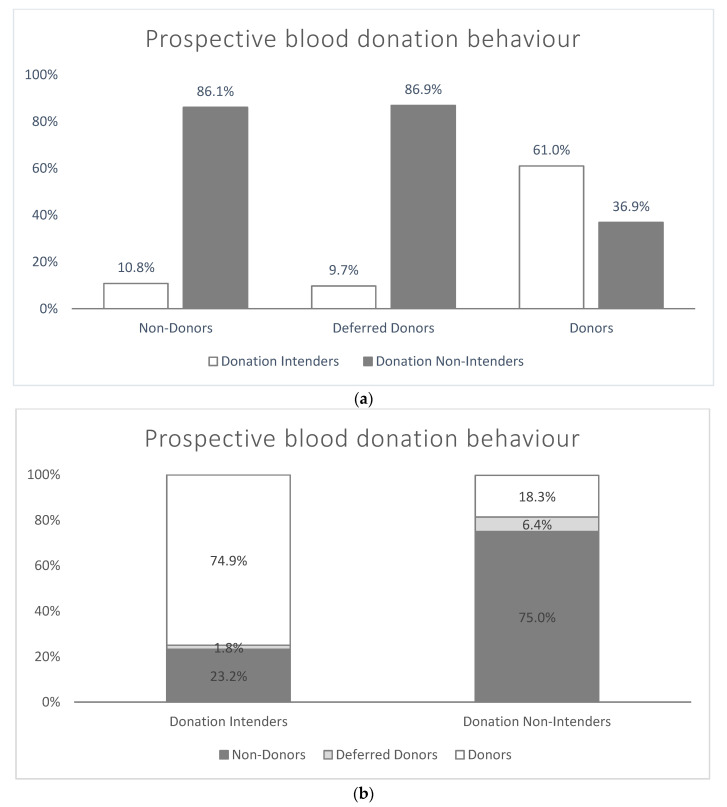
(**a**) Self-reported prospective (12 months) blood donation behaviour (aggregated) depending on blood donation experiences—results for total sample (*n* = 2531, weighted). (**b**) Proportions for different types of blood donation experiences as related to prospective (12 months) blood donation intentions—results for total sample (*n* = 2531, weighted). Absolute frequencies and cumulative percentages vary as a function of the amount of missing data for each variable.

**Table 1 ijerph-18-04238-t001:** Sociodemographic characteristics of study sample (*n* = 2531, weighted) *.

	Total ***		Non- Donors		Deferred Donors		Donors		Donation Non-Intenders		Donation- Intenders	
	*n*	*%*	*n*	*%*	*n*	*%*	*n*	*%*	*n*	*%*	*n*	*%*
**Sex**												
Male	**1241**	***49.0***	716	*46.9*	50	*38.7*	474	*54.5*	825	*47.1*	382	*53.9*
Female	**1290**	***51.0***	810	*53.1*	79	*61.3*	396	*45.5*	927	*52.9*	326	*46.1*
**Age Group**												
less than 18 years	**95**	***3.8***	86	*35.6*	3	*2.0*	7	*0.8*	26	*4.3*	19	*2.7*
18–24 years	**249**	***9.8***	178	*11.7*	8	*6.2*	62	*7.1*	*127*	*7.2*	112	*15.9*
25–34 years	**356**	***14.0***	212	*13.9*	21	*16.0*	121	*13.9*	217	*12.4*	131	*18.5*
35–44 years	**357**	***14.1***	202	*13.2*	20	*15.2*	135	*15.5*	223	*12.7*	118	*16.7*
45–54 years	**458**	***18.1***	251	*16.4*	24	*18.2*	183	*21.2*	282	*16.1*	164	*23.2*
55–64 years	**398**	***15.7***	231	*15.1*	22	*16.6*	146	*16.7*	290	*16.5*	98	*13.8*
65–74 years	**362**	***14.3***	208	*13.6*	24	*18.5*	131	*15.0*	308	*17.6*	47	*6.7*
75–84 years	**223**	***8.8***	139	*9.1*	9	*6.6*	75	*8.7*	202	*11.5*	16	*2.3*
85 years or older	**33**	***1.3***	20	*1.3*	1	*0.7*	10	*1.1*	29	*1.7*	1	*0.2*
**Nationality ****												
German	**2425**	***95.8***	1435	*94.1*	127	*98.0*	859	*98.8*	1671	*95.4*	693	*97.8*
Other	**106**	***4.2***	91	*5.9*	3	*2.0*	11	*1.2*	80	*4.6*	15	*2.2*
**Confession**												
Protestant	**940**	***37.1***	582	*38.1*	44	*33.6*	314	*36.1*	653	*37.3*	262	*36.9*
Catholic	**788**	***31.1***	473	*31.0*	43	*33.0*	268	*30.8*	549	*31.3*	222	*31.4*
Muslim	**67**	***2.6***	59	*3.9*	1	*0.8*	5	*0.6*	48	*2.8*	10	*1.4*
Other	**53**	***2.1***	28	*1.8*	5	*3.9*	20	*2.3*	32	*1.8*	21	*2.9*
None	**584**	***23.1***	323	*21.2*	32	*24.4*	228	*26.2*	399	*22.8*	168	*23.8*
**Family Status**												
Married (living together)	**1194**	***47.2***	681	*44.7*	66	*51.3*	445	*51.2*	857	*48.9*	305	*43.0*
Married (living departed)	**30**	***1.2***	18	*1.2*	1	*0.8*	8	*0.9*	20	*1.2*	8	*1.1*
Living with unmarried partner	**319**	***12.6***	181	*11.8*	21	*16.5*	117	*13.4*	197	*11.3*	113	*15.9*
Single (living without partner)	**591**	***23.4***	388	*25.4*	20	*15.5*	176	*20.2*	360	*20.6*	214	*30.2*
Divorced (living without partner)	**194**	***7.7***	126	*8.3*	3	*2.3*	62	*7.1*	134	*7.7*	54	*7.6*
Widowed (living without partner)	**196**	***7.7***	122	*8.0*	14	*10.9*	55	*6.3*	177	*10.1*	14	*2.0*
**Education (Qualification)**												
No formal qualification	**53**	***2.1***	43	*2.8*	1	*1.1*	8	*0.9*	41	*2.4*	7	*1.0*
Lowest formal qualification (8–9 years)	**767**	***30.3***	492	*32.2*	36	*27.5*	237	*27.2*	618	*35.3*	134	*18.9*
Intermediary secondary qualification (10 years)	**977**	***38.6***	555	*36.3*	51	*39.3*	370	*42.6*	619	*35.4*	328	*46.2*
Higher secondary education qualification (11–13 years)	**404**	***15.9***	231	*15.1*	18	*14.4*	153	*17.6*	242	*13.8*	148	*20.9*
University degree	**223**	***8.8***	114	*7.4*	19	*15.0*	90	*10.4*	146	*8.3*	70	*9.9*
Any other formal degree	**5**	***0.2***	2	*0.2*	-	*-*	3	*0.3*	5	*0.3*	-	*-*
Still in formal school education (no degree yet)	**93**	***3.7***	87	*5.7*	3	*2.0*	3	*0.4*	74	*4.2*	18	*2.6*
**Income (Household)**												
Up to 1500 Euro	**496**	***19.6***	323	*21.1*	23	*17.5*	149	*17.1*	373	*21.3*	102	*14.3*
Up to 2500 Euro	**768**	***30.3***	447	*29.3*	37	*28.2*	282	*32.4*	554	*31.6*	199	*28.0*
Up to 3500 Euro	**599**	***23.7***	355	*23.3*	34	*26.6*	209	*24.0*	403	*23.0*	181	*25.5*
More than 3500 Euro	**565**	***22.3***	329	*21.6*	29	*22.1*	207	23.8	348	*19.9*	202	*28.5*

* Absolute frequencies and cumulative percentages vary as a function of the amount of missing data for each variable. ** Categories are not mutually exclusive (due to double citizenship). *** In **bold** print: frequencies and percentages for total sample.

**Table 2 ijerph-18-04238-t002:** (**a**) Socio-demographic characteristic of past non-donors, deferred donors, or donors (*n* = 2531, weighted) *. (**b**) Socio-demographic characteristic of non-intenders and intenders (*n* = 2531, weighted) *.

(a)								
	Non-Donors (Previous Lifetime)		Deferred Donors (Previous Lifetime)		Donors (Previous Lifetime)		Effect Size (Sig.)	
	*n*	*%*	*n*	*%*	*n*	*%*	Cramer’s V (*p*-Value)	
**Total**	1526	60.4	130	5.1	870	34.4		
**Sex**							0.086	0.000
Male	716	46.9	50	38.7	474	54.5		
Female	810	53.1	79	61.3	396	45.5		
**Age Group**							0.111	0.000
less than 18 years	86	5.6	3	2.0	7	0.8		
18–24 years	178	11.7	8	6.2	62	7.1		
25–34 years	212	13.9	21	16.0	121	13.9		
35–44 years	202	13.2	20	15.2	135	15.5		
45–54 years	251	16.4	24	18.2	183	21.1		
55–64 years	231	15.1	22	16.6	146	16.7		
65–74 years	208	13.6	24	18.5	131	15.0		
75–84 years	139	9.1	9	6.6	75	8.7		
85 years or older	20	1.3	1	0.7	10	1.1		
**Confession**							0.084	0.000
Protestant	582	38.1	44	33.6	314	36.1		
Catholic	473	31.0	43	33.0	268	30.8		
Muslim	59	3.9	1	0.8	5	0.6		
Other	28	1.8	5	3.9	20	2.3		
None	323	21.2	32	24.4	228	26.2		
**Family Status**							0.069	0.020
Married (living together)	681	44.7	66	51.3	445	51.2		
Married (living departed)	39	2.6	4	3.1	16	1.8		
Living with unmarried partner	112	7.4	15	11.7	73	8.4		
Single (living without partner)	388	25.4	20	15.1	176	20.2		
Divorced (living without partner)	160	10.5	10	7.7	90	10.4		
Widowed (living without partner)	136	8.9	14	11.1	65	7.4		
**Education (Qualification)**							0.127	0.000
No formal qualification	43	2.8	1	1.1	8	0.9		
Lowest formal qualification (8–9 years)	492	32.2	36	27.5	237	27.2		
Intermediary secondary qualification (10 years)	553	36.3	51	39.3	370	42.6		
Higher secondary education qualification (11–13 years)	231	15.1	19	14.4	153	17.6		
University degree	114	7.4	19	15.0	90	10.4		
Any other formal degree	2	0.2	0	0.0	3	0.3		
Still in formal school education (no degree yet)	87	5.7	3	2.0	3	0.4		
**Income (Household)**							0.054	0.021
Up to 1500 Euro	323	21.1	23	17.5	149	17.1		
Up to 2500 Euro	802	52.6	71	54.8	491	56.5		
Up to 3500 Euro	329	21.6	29	22.1	207	23.8		
**(b)**								
	**Non-Intenders** **(Next 12 Months)**		**Intenders** **(Next 12 Months)**				**Effect** **Size** **(Sig.)**	
	***n***	***%***	***n***		***%***		**Cramer’s V** **(*p*-Value)**	
**Total**	1752	71.2	708		28.8			
**Sex**							0.062	0.002
Male	825	47.1	382		53.9			
Female	927	52.9	326		46.1			
**Age Group**							0.268	0.000
less than 18 years	75	4.3	19		2.7			
18–24 years	127	7.2	112		15.9			
25–34 years	217	12.4	131		18.5			
35–44 years	223	12.7	118		16.7			
45–54 years	282	16.1	164		23.2			
55–64 years	290	16.5	98		13.8			
65–74 years	308	17.6	47		6.7			
75–84 years	202	11.5	16		2.3			
85 years or older	29	1.7	1		0.2			
**Confession**							0.055	0.195
Protestant	653	37.3	262		36.9			
Catholic	549	31.3	222		31.4			
Muslim	48	2.8	10		1.4			
Other	32	1.8	21		2.9			
None	399	22.8	168		23.8			
**Family Status**							0.183	0.000
Married (living together)	857	48.9	305		43.0			
Married (living departed)	42	2.4	13		1.8			
Living with unmarried partner	114	6.5	82		11.5			
Single (living without partner)	357	20.4	210		29.6			
Divorced (living without partner)	176	10.0	78		11.1			
Widowed (living without partner)	193	11.0	19		2.6			
**(Qualification)**							0.189	0.000
No formal qualification	41	2.4	7		1.0			
Lowest formal qualification (8–9 years)	618	35.3	134		18.9			
Intermediary secondary qualification (10 years)	619	35.3	328		46.2			
Higher secondary education qualification (11–13 years)	242	13.8	148		20.9			
University degree	146	8.3	70		9.9			
Any other formal degree	5	0.3	0		0.0			
Still in formal school education (no degree yet)	74	4.2	18		2.6			
**Income (Household)**							0.110	0.000
Up to 1500 Euro	373	21.3	102		14.3			
Up to 2500 Euro	957	54.6	379		53.5			
Up to 3500 Euro	348	19.9	202		28.5			

* Absolute frequencies and cumulative percentages vary as a function of the amount of missing data for each variable. ** Effect size interpretation: Cramer’s V > 0.10 = small effect; Phi > 0.30 = medium effect; Phi > 0.50 = large effect (Cohen, 1988).

**Table 3 ijerph-18-04238-t003:** Category system for self-reported blood donation motivators and deterrents differentiated by domains, main categories, and subcategories, including definitions and anchoring items.

Main Categories	Subcategories	Definition	Anchor Example
**Ineligibility**		**All statements relating to reasons for a missing/inadequate “suitability” for blood donation (e.g., existing diseases)**	
Unspecific reason		Lack of “eligibility” without giving specific reasons.	“I am not suitable.”
Specific reason	Age	Lack of “eligibility” because individual is too young or old to donate blood.	“Due to age.”
	Health	Lack of “eligibility” due to health restrictions (e.g., illness).	“Not allowed because of illness.”
	Pregnancy	Temporary lack of “eligibility” due to pregnancy.	“I am pregnant.”
Specific reason	other	Statements of other specific reasons.	“Trips abroad”
**Impact and Effect**		**Statements concerning the anticipated impact/effect**	
Physical consequences	Positive	Due to the expected positive physical effects for the donor (e.g., vitality).	“I’m doing something good for myself, my body.”
	Negative—health	Due to the expected negative physical effects for the donor (e.g., cardiovascular system problems).	“My cardiovascular system can’t really handle it”
	Negative—risk	Due to the expected health risks for the donor (e.g., infection).	“Because of the risk of infection.”
Mental/psychological well-being	Positive	Due to the expected positive psychological effects for the donor (e.g., wellbeing).	“It’s good for me.”
	Negative	Due to the expected negative psychological effects for the donor (e.g., feeling unwell).	“It’s not good for me”
**Fears and Aversion**		**All statements relating to possible aversions and fears regarding blood donation.**	
Fears	Fears in general	All statements relating to fear in general, without further specification.	“Am afraid.”
	Fear of the needle	Due to existing/anticipated fear of needles.	“Fear of needles.”
	Fear of blood	Due to existing/anticipated fear of blood.	“Am afraid of blood.”
	Fear to donate	Due to existing/anticipated fear to donate.	“Am afraid of blood getting taken.”
	Fear of pain	Due to the existing/anticipated fear of pain.	“Fear of pain.”
	other	Statements relating to specific other fears.	“Fear of doctors.”
Aversion	Needles	Due to a personal aversion to needles.	“Have an aversion to needles and syringes.”
	Blood	Due to a personal aversion to blood.	“Can’t see blood.”
	other	Statements relating to other specific aversions.	“I dislike it.”
**Obstacles and Barriers**		**All statements relating to possible logistical/organizational obstacles.**	
Lack of information		Due to a lack of information and knowledge about blood donation.	“I wouldn’t even know where and how I could donate.”
No opportunity/Lack of possibilities		Due to a lack of opportunities/possibilities (e.g., distance to blood donation)	“No opportunity nearby.”
Organization/effort		Due to the organizational effort involved.	“Too cumbersome, complicated.”
Time/Lack of time		Due to a lack of time.	“I hardly have time for that.”
Personal reasons		Due to personal reasons regarding logistics/organization but that do not fall under the categories named above.	“Can’t set it up.”
**Norms**		**All statements relating to normative reasons/motives.**	
Reciprocity	General (if none of the three subcategories)	Due to the recognition of the norm of reciprocity because of past or potential future utilization.	“Blood for blood.”
	Future-orientated	Due to the expectation of increasing the future possibility to receive someone else’s blood by donating blood now.	“A situation could come up in which one needs blood.”
	Past-orientated (self)	Due to the recognition of the norm of reciprocity because of having received someone else’s blood in the past.	“To give back something of what I have received.”
	Past-orientated (friends and family)	Due to the recognition of the norm of reciprocity because friends/family having received someone else’s blood in the past.	“In return for the blood my child received.”
Altruism		Due to the unconditional necessity of helping people (in need) and saving lives.	“That is a way in which I can save lives.”
(Feeling of) Obligation/self-evident/social conscientiousness and responsibility		Generally described subjective obligation of the social norm/expectation to donate blood.	“To do something for the community.”
Religious reasons		Due to religious/denominational reasons.	“For religious reasons.”
Personal belief		Due to personal beliefs, if not described through another normative belief.	“It is important for me personally.”
Important/necessary/meaningful/good cause		Generally described relevance to donate blood.	“Because it is important.”
Vocational affiliation		Due to one’s own vocation (e.g., working in the health care system).	“Through my work in the clinic.”
**Main categories**	**Subcategories**	**Definition**	**Anchor example**
Surrendering responsibility		Due to the non-existent insight into why individuals should become active/Personal need of one’s blood.	“Enough others do it.”/“I need my blood myself.”
**Image and Experience**		**All motives that relate to a specific characteristic of blood donation.**	
Compensate for the lack of blood products/Support the health care system		Due to the knowledge of the lack of blood conserves and the necessity of blood donation for a functioning health care system	“Demand is constantly rising.”
Lack of trust (rip-off/crime/fraud/profiteering—media reports)		Due to a lack of trust in the blood donation system, amongst others formulated in the form of general charges	“Because I don’t want to support such profiteering.”
Try it out (curiosity motive)		Curiosity	“I was curious.”
No need present		Due to a lack of need	“There is no lack of blood.”
Rare blood type		Due to the knowledge of the rarity (and thus the relevance of the donation) of the own blood type	“Important, I have a rare blood type.”
Advertisement/Campaign/Phone call/Appeal		Donated due to campaigns or advertisements	“Promo day.”
(Missing) previous experience or habit		Due to already collected experience in donating	“I am a permanent donor and regularly donate blood.”
Absence of disadvantages		Due to the absence of obstacles/disadvantages	“Nothing bad” “it doesn’t hurt me”
**Benefits and Incentives**		**All motives mentioning compensatory measures.**	
Blood donor card		Receive a blood donor card	“Interested in a blood-type card.”
Compensation		Financial compensation	“Money!”
Determination of the blood type		Determination of the blood type free of charge	“That way I could learn my blood type.”
Health check/screening		Health check/screening free of charge	“At the same time one gets a health check.”
Exemption from work/school		Exemption from work/school to donate blood	“I wanted a little time off work.”
Other services		Due to services/discounts that the donor receives	“Extra holiday” “…and there was pea soup”
**Conditions**		**Statements of specific conditions in which blood donations would be given.**	
For personal need only		For personal treatment/prevention only	“For myself alone.”
Only for family/important others/if person is known		Blood donation for close relatives or acquaintances/trusted persons only	“Why should I? If, then only for relatives.”
In case of a disaster/emergency/personal experience		Due to special demand or circumstance	“Car crash of my parents” “Train accident”
**Psychological Aspects**		**Statements related to aspects of attitude, volition, and behaviour.**	
Will be made up for/is planned		Due to the fact that the respondent has not donated blood despite existing intention, but plans to make up for this	“I want to do it soon.”
Not ready yet		Due to the fact that the respondent doesn’t feel ready to donate blood	“I’m not ready for it yet.”
No interest/no will		Due to the fact that the respondent is unwilling and uninterested to donate blood.	“It doesn’t interest me.”
Indifference/passivity/negligence/comfort/no desire		Due to the fact that the respondent is indifferent/passive concerning blood donation or negligent/desireless or too comfortable to donate	“I don’t desire to do it.”
Not thought about it		Due to the fact that the respondent has not thought about it yet	“Haven’t thought about it yet.”
Social aspect/peer group movement/personal influence and advice		Due to social motivation	“My ex-girlfriend took me there.”
Refusal to donate blood		Due to the belief that blood donation isn’t important/meaningless or is refused	“Meaningless”
**Missing points of contact**		**Statements related to missing triggers/contact points.**	
No request/appellation/call		Due to the fact that the respondent has not explicitly been asked/called upon to do so or been addressed separately	“Someone should ask me about it once.”
Missing reason/occasion		Due to the fact that there has not been a special occasion or specific reason to do so	“There hasn’t been an occasion to do it yet.”
For no reason		Lack of awareness of the motivations and obstacles	“No reason.”
**Misc**		**Other statements**	
Other		Due to aspects that cannot be assigned to the other categories	(various)
Do not know		Respondent doesn’t know/cannot remember anymore	“Don’t know.”
Not specified		Statement is missing or was actively refused	“None.”

**Table 4 ijerph-18-04238-t004:** (**a**) Self-reported motivators and deterrents for retrospective (lifetime) blood donation behaviour—results for different donor types (*n* = 2531. unweighted) ***/**/***/******. (**b**) Self-reported motivators and deterrents for prospective (12 months) blood donation behaviour—results for different donor types (*n* = 2531, unweighted) ***/**/***/******.

(a)								
Domain (D)/Main Category	Subcategory	Non-Donors (Previous Lifetime)		Deferred Donors (Previous Lifetime)		Donors (Previous Lifetime)		Effect Size (Sig.)
D01 Ineligibility		*n*	*%*	*n*	*%*	*n*	*%*	*Cramer’s V* *(p-Value)*
Unspecific reason		12	*0.8*	3	*2.3*	0	*0.0*	0.071 (0.002)
Specific reason	Age	**110**	***7.3***	5	*3.8*	7	*0.8*	0.144 (0.000)
	Health	**207**	***13.8***	**75**	***56.4***	19	*2.1*	0.365 (0.000)
	Pregnancy	6	*0.4*	0	*0.0*	0	*0.0*	0.040 (0.128)
Specific reason	Other	12	*0.8*	8	*6.0*	3	*0.3*	0.129 (0.000)
**D02 Impact/Effect**	***Total n/domain (D01)***	**336**	***22.4***	**90**	**67.7**	*27*	*3.0*	0.387 (0.000)
Physical consequences	Positive	0	*0.0*	1	*0.8*	19	*2.1*	0.113 (0.000)
	Negative—health	67	*4.5*	1	*0.8*	4	*0.4*	0.118 (0.000)
	Negative—risk	36	*2.4*	0	*0.0*	1	*0.1*	0.094 (0.000)
Mental/psychological wellbeing	Positive	1	*0.1*	0	*0.0*	16	*1.8*	0.101 (0.000)
	Negative	7	*0.5*	0	*0.0*	0	*0.0*	0.044 (0.091)
**D03 Fears/Aversion**	***Total n/domain (D02)***	**105**	***7.0***	2	*1.6*	37	*4.1*	0.072 (0.001)
Fears	Fears in general	43	*2.9*	0	*0.0*	0	*0.0*	0.109 (0.000)
	Fear of the needle	**90**	***6.0***	0	*0.0*	0	*0.0*	0.159 (0.000)
	Fear of blood	5	*0.3*	0	*0.0*	0	*0.0*	0.037 (0.180)
	Fear to donate	14	*0.9*	0	*0.0*	0	*0.0*	0.062 (0.008)
	Fear of pain	3	*0.2*	0	*0.0*	0	*0.0*	0.029 (0.358)
	Other	9	*0.6*	1	*0.8*	0	*0.0*	0.047 (0.062)
Aversion	Needles	28	*1.9*	0	*0.0*	0	*0.0*	0.088 (0.000)
	Blood	35	*2.3*	0	*0.0*	0	*0.0*	0.098 (0.000)
	Other	23	*1.5*	0	*0.0*	0	*0.0*	0.079 (0.000)
**D04 Obstacles/Barriers**	***Total n/domain (D03)***	**219**	***14.6***	1	*0.8*	**0**	**0.0**	0.253 (0.000)
Lack of information		32	*2.1*	1	*0.8*	1	*0.1*	0.084 (0.000)
No opportunity/Lack of possibilities		**94**	***6.3***	0	*0.0*	3	*0.3*	0.153 (0.000)
Organization/effort		15	*1.0*	1	*0.8*	0	*0.0*	0.059 (0.011)
Time/Lack of time		**137**	***9.1***	2	*1.5*	4	*0.4*	0.182 (0.000)
Personal reasons		46	*3.1*	0	*0.0*	1	*0.1*	0.108 (0.000)
**D05 Norms**	***Total n/domain (D04)***	**292**	***19.5***	3	*2.3*	8	0.9	0.278 (0.000)
Reciprocity	General (if no other category)	1	*0.1*	1	*0.8*	6	*0.7*	0.054 (0.026)
	Future-orientated	1	*0.1*	3	*2.3*	**51**	***5.7***	0.182 (0.000)
	Past-orientated (self)	1	*0.1*	2	*1.5*	10	*1.1*	0.077 (0.001)
	Past-orientated (friends and family)	2	*0.2*	0	*0.0*	5	*0.5*	0.040 (0.130)
Altruism		16	*1.1*	**18**	***13.5***	**405**	***45.4***	0.551 (0.000)
Obligation/self-evident/social conscientiousness and responsibility		4	*0.3*	0	*0.0*	**109**	***12.2***	0.277 (0.000)
Religious reasons		10	*0.7*	0	*0.0*	3	*0.3*	0.028 (0.382)
Personal belief		0	*0.0*	1	*0.8*	27	*3.0*	0.136 (0.000)
Important/necessary/meaningful/good cause		7	*0.5*	3	*2.3*	**84**	***9.4***	0.223 (0.000)
Vocational affiliation		1	*0.1*	1	*0.8*	28	*3.1*	0.134 (0.000)
Surrendering responsibility		15	*1.0*	0	*0.0*	0	*0.0*	0.064 (0.006)
	***Total n/domain (D05)***	57	*3.8*	**26**	***19.5***	**633**	**70.9**	0.702 (0.000)
**D06** **Image/Experience**		***n***	***%***	***n***	***%***	***n***	***%***	***Cramer’s V*** ***(p-Value)***
Compensate for lack of blood products/Support the health care system		2	*0.1*	3	*2.3*	**64**	***7.2***	0.203 (0.000)
Lack of trust (rip-off/crime/fraud/profiteering—media reports)		31	*2.1*	0	*0.0*	0	*0.0*	0.092 (0.000)
Try it out (curiosity motive)		0	*0.0*	0	*0.0*	4	*0.4*	0.054 (0.026)
No need present		6	*0.4*	0	*0.0*	0	*0.0*	0.040 (0.128)
Rare blood type		0	*0.0*	1	*0.8*	14	*1.6*	0.096 (0.000)
Advertisement/Campaign/Phone call/Appeal		0	*0.0*	2	*1.5*	26	*2.9*	0.131 (0.000)
(Missing) previous experience or habit		11	*0.7*	0	*0.0*	**45**	***5.0***	0.142 (0.000)
Absence of disadvantages		2	*0.1*	1	*0.8*	17	*1.9*	0.094 (0.000)
**D07 Benefits/Incentives**	***Total n/domain (D06)***	52	*3.4*	7	*5.3*	**164**	***18.4***	0.249 (0.000)
Blood donor card		0	*0.0*	0	*0.0*	3	*0.3*	0.047 (0.064)
Compensation		3	*0.2*	2	*1.5*	**61**	***6.8***	0.196 (0.000)
Determination of the blood type		0	*0.0*	0	*0.0*	14	*1.6*	0.101 (0.000)
Health check/screening		0	*0.0*	0	*0.0*	21	*2.4*	0.124 (0.000)
Exemption from work/school		0	*0.0*	0	*0.0*	9	*1.0*	0.081 (0.000)
Other services		0	*0.0*	0	*0.0*	14	*1.6*	0.101 (0.000)
**D08 Conditions**	***Total n/domain (D07)***	3	*0.2*	2	*1.5*	**107**	***12.0***	0.271 (0.000)
For personal need only		4	*0.3*	0	*0.0*	4	*0.4*	0.020 (0.598)
Only for family/important others/if person is known		7	*0.5*	0	*0.0*	3	*0.3*	0.018 (0.670)
In case of a disaster/emergency/personal experience		5	*0.3*	0	*0.0*	9	*1.0*	0.046 (0.067)
**D09 Psychological Aspects**	***Total n/domain (D08)***	16	*1.1*	0	*0.0*	16	*1.7*	0.041 (0.126)
Will be made up for/is planned		23	*1.5*	1	*0.8*	0	*0.0*	0.075 (0.001)
Not ready yet		18	*1.2*	1	*0.8*	0	*0.0*	0.065 (0.005)
No interest/no will		**126**	***8.4***	1	*0.8*	0	*0.0*	0.187 (0.000)
Indifference/passivity/negligence/comfort/no desire		53	*3.5*	0	*0.0*	1	*0.1*	0.117 (0.000)
Not thought about it		**161**	***10.7***	0	*0.0*	0	*0.0*	0.216 (0.000)
Social aspect/peer group movement/personal influence and advice		3	*0.2*	0	*0.0*	42	*4.7*	0.163 (0.000)
Refusal to donate blood		10	*0.7*	0	*0.0*	0	*0.0*	0.052 (0.032)
**D10 Missing points of contact**	***Total n/domain (D09)***	**386**	***25.8***	3	*2.4*	43	*4.8*	0.278 (0.000)
No request/appellation/call		26	*1.7*	0	*0.0*	1	*0.1*	0.078 (0.000)
Missing reason/occasion		18	*1.2*	0	*0.0*	0	*0.0*	0.070 (0.002)
For no reason		23	*1.5*	0	*0.0*	5	*0.6*	0.050 (0.040)
**Misc**	***Total n/domain (D10)***	67	*4.4*	0	*0.0*	6	*0.7*	0.114 (0.000)
Other		13	*0.9*	1	*0.8*	11	*1.2*	0.018 (0.657)
Do not know		24	*1.6*	0	*0.0*	5	*0.6*	0.053 (0.031)
Not specified		**162**	***10.8***	**16**	***12.0***	29	*3.2*	0.134 (0.000)
	***Total n/domain (D11)***	**199**	***13.3***	**17**	***12.8***	**45**	***5.0***	0.129 (0.000)
**(b)**								
**Domain (D)/Main Category**	**Subcategory**	**Non-Intenders** **(Next 12 Months)**			**Intenders** **(Next 12 Months)**		**Effect** **Size**	
**D01 Ineligibility**		***n***	***%***		***n***	***%***	***Phi***	
Unspecific reason		20	*1.2*		0	*0.0*	−0.059 (0.003)	
Specific reason	Age	**284**	***16.5***		5	*0.7*	−0.225 (0.000)	
	Health	**337**	***19.6***		3	*0.4*	−0.254 (0.000)	
	Pregnancy	7	*0.4*		0	*0.0*	−0.035 (0.083)	
Specific reason	Other	14	*0.8*		1	*0.1*	−0.040 (0.049)	
**D02 Impact/Effect**	***Total n/domain (D01)***	**634**	***36.8***		9	*1.2*	−0.371 (0.000)	
Physical consequences	Positive	2	*0.1*		12	*1.6*	0.092 (0.000)	
	Negative—health	62	*3.6*		2	*0.3*	−0.096 (0.000)	
	Negative—risk	29	*1.7*		0	*0.0*	−0.071 (0.000)	
Mental/psychological wellbeing	Positive	0	*0.0*		19	*2.6*	0.135 (0.000)	
	Negative	6	*0.3*		0	*0.0*	−0.032 (0.109)	
**D03 Fears/Aversion**	***ToTotal n/domain (D02)***	**95**	***5.5***		31	*4.2*	−0.027 (0.181)	
Fears	Fears in general	40	*2.3*		0	*0.0*	−0.084 (0.000)	
	Fear of the needle	80	*4.6*		0	*0.0*	−0.120 (0.000)	
	Fear of blood	5	*0.3*		0	*0.0*	−0.030 (0.144)	
	Fear to donate	20	*1.2*		0	*0.0*	−0.059 (0.003)	
	Fear of pain	2	*0.1*		0	*0.0*	−0.019 (0.355)	
	Other	11	*0.6*		1	*0.1*	−0.033 (0.102)	
Aversion	Needles	21	*1.2*		0	*0.0*	−0.061 (0.003)	
	Blood	27	*1.6*		0	*0.0*	−0.069 (0.001)	
	Other	21	*1.2*		2	*0.3*	−0.045 (0.026)	
**D04 Obstacles/Barriers**	***Total n/domain (D03)***	**205**	***11.9***		3	*0.4*	−0.189 (0.000)	
Lack of information		27	*1.6*		5	*0.7*	−0.036 (0.076)	
No opportunity/Lack of possibilities		18	*1.0*		10	*1.4*	0.014 (0.500)	
Organization/effort		27	*1.6*		0	*0.0*	−0.069 (0.001)	
Time/Lack of time		**128**	***7.4***		20	*2.7*	−0.091 (0.000)	
Personal reasons		33	*1.9*		3	*0.4*	−0.057 (0.004)	
**D05 Norms**	***Total n/domain (D04)***	**209**	***12.1***		**36**	***4.9***	−0.111 (0.000)	
Reciprocity	General (if no other category)	2	*0.1*		4	*0.5*	0.040 (0.049)	
	Future-orientated	8	*0.5*		**51**	***6.9***	0.194 (0.000)	
	Past-orientated (self)	1	*0.1*		8	*1.1*	0.078 (0.000)	
	Past-orientated (friends and family)	0	*0.0*		1	*0.1*	0.031 (0.126)	
Altruism		24	*1.4*		**297**	***40.4***	0.530 (0.000)	
Obligation/self-evident/social conscientiousness and responsibility		7	*0.4*		**69**	***9.4***	0.238 (0.000)	
Religious reasons		7	*0.4*		1	*0.1*	−0.022 (0.281)	
Personal belief		4	*0.2*		16	*2.2*	0.099 (0.000)	
Important/necessary/meaningful/good cause		3	*0.2*		**77**	***10.5***	0.266 (0.000)	
Vocational affiliation		0	*0.0*		8	*1.1*	0.087 (0.000)	
Surrendering responsibility		25	*1.5*		0	*0.0*	−0.066 (0.001)	
	***Total n/domain (D05)***	77	*4.5*		**474**	***64.5***	0.659 (0.000)	
**D06** **Image and Experience**		***n***	***%***		***n***	***%***	***Phi***	
Compensate for lack of blood products /Support the health care system		2	*0.1*		**68**	***9.3***	0.251 (0.000)	
Lack of trust (rip-off/crime/fraud/profiteering—media reports)		34	*2.0*		3	*0.4*	−0.059 (0.004)	
Try it out (curiosity motive)		0	*0.0*		2	*0.3*	0.044 (0.030)	
No need present		8	*0.5*		1	*0.1*	−0.025 (0.217)	
Rare blood type		2	*0.1*		11	*1.5*	0.087 (0.000)	
Advertisement/Campaign/Phone call/Appeal		3	*0.2*		4	*0.5*	0.032 (0.115)	
(Missing) previous experience or habit		16	*0.9*		**38**	***5.2***	0.132 (0.000)	
Absence of disadvantages		3	*0.2*		30	*4.1*	0.155 (0.000)	
**D07 Benefits/Incentives**	***Total n/domain (D06)***	68	*4.0*		**149**	***20.3***	0.263 (0.000)	
Blood donor card		0	*0.0*		0	*0.0*	*-*	
Compensation		19	*1.1*		19	*2.6*	0.055 (0.006)	
Determination of the blood type		0	*0.0*		1	*0.1*	0.031 (0.126)	
Health check/screening		0	*0.0*		11	*1.5*	0.103 (0.000)	
Exemption from work/school		0	*0.0*		0	*0.0*	*-*	
Other services		3	*0.2*		5	*0.7*	0.041 (0.044)	
**D08 Conditions**	***Total n/domain (D07)***	22	*1.3*		33	*4.5*	0.099 (0.000)	
For personal need only		5	*0.3*		2	*0.3*	−0.002 (0.938)	
Only for family/important others/if person is known		11	*0.6*		5	*0.7*	0.002 (0.907)	
In case of a disaster/emergency/personal experience		8	*0.5*		4	*0.5*	0.005 (0.795)	
**D09 Psychological Aspects**	***Total n/domain (D08)***	22	*1.3*		11	*1.5*	0.009 (0.666)	
Will be made up for/is planned		7	*0.4*		29	*3.9*	0.135 (0.000)	
Not ready yet		17	*1.0*		2	*0.3*	−0.037 (0.064)	
No interest/no will		**117**	***6.8***		4	*0.5*	−0.132 (0.000)	
Indifference/passivity/negligence/comfort/no desire		36	*2.1*		3	*0.4*	−0.062 (0.002)	
Not thought about it		42	*2.4*		5	*0.7*	−0.059 (0.004)	
Social aspect/peer group movement/personal influence and advice		5	*0.3*		18	*2.4*	0.103 (0.000)	
Refusal to donate blood		8	*0.5*		0	*0.0*	−0.037 (0.064)	
**D10 Missing points of contact**	***Total n/domain (D09)***	**229**	***13.3***		**59**	***8.0***	−0.075 (0.000)	
No request/appellation/call		11	*0.6*		4	*0.5*	−0.006 (0.783)	
Missing reason/occasion		15	*0.9*		1	*0.1*	−0.042 (0.038)	
For no reason		19	*1.1*		3	*0.4*	−0.034 (0.094)	
**Misc**	***Total n/domain (D10)***	44	*2.6*		8	*1.0*	−0.047 (0.021)	
Other		16	*0.9*		9	*1.2*	0.013 (0.504)	
Do not know		26	*1.5*		3	*0.4*	−0.047 (0.021)	
Not specified		**241**	***14.0***		**48**	***6.5***	−0.106 (0.000)	
	***Total n/domain (D11)***	**283**	***16.4***		**60**	***8.1***	−0.109 (0.000)	

(**a**): * Categories are not mutually exclusive. ** In **bold** print: category/domain percentages >5%. ***** Total number per domain indicates number of cases with at least one category in this domain. **** Effect size interpretation: Cramer’s V > 0.10 = small effect; Phi > 0.30 = medium effect; Phi > 0.50 = large effect (Cohen, 1988); (**b**): * Categories are not mutually exclusive. ** In **bold** print: category/domain percentages >5%. ***** Total number per domain indicates number of cases with at least one category in this domain. **** Effect size interpretation: Phi > 0.10 = small effect; Phi > 0.30 = medium effect; Phi > 0.50 = large effect (Cohen, 1988).

## Data Availability

The datasets generated and/or analysed during the current study are available from the corresponding author on reasonable request.

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
