# Peer review of "“Blood for Blood”? Personal Motives and Deterrents for Blood Donation in the German Population"

_ijerph, 2021, doi:10.3390/ijerph18084238_

Round 1
Reviewer 1 Report
The study under review examines a very interesting and important topic. I believe that this study provides valuable information that both academics and practitioners would like to acquire. However, the current version study is not without problems. Below, I outline a number of places that have room for improvement.
First, the paper claims that the study sample is representative of the German population. But the description of the sampling procedure is far from adequate. From nowhere can the reader understand why this sample is representative of the German population. More details on the sampling and the survey procedure should be provided to better inform the reader.
Second, I can see the research team devoted considerable efforts to validating data coding. However, it would be a pity not to dig the data a little deeper given this effort. To be specific, the analysis done throughout the paper is basically descriptive in nature, but I think there is much room for more sophisticated statical analyses. For example, who (e.g., people of a certain age, gender, and education level, etc.) are more likely to have the motivators and who are more likely to have deferents of blood donation? You can run some regressions to answer questions like this.
Some simple t-tests would also be useful. For example, Table 4 shows the differences in the values of some variables between intenders and non-intenders, but it would be more informative if you show people whether these differences are statistically different across the two types of people. Without a (statistical) benchmark, it's hard to understand whether 5% is a large number or a negligible number in your context.
Finally, while the paper is in general well-written, I found grammatical errors at times. There are also a lot of inconsistencies in citation style. The authors should check the language more carefully to improve the quality of communication.
Author Response
Dear Reviewer 1, thank you very much for reviewing our manuscript. We appreciated your comments and address them below:
First, the paper claims that the study sample is representative of the German population. But the description of the sampling procedure is far from adequate. From nowhere can the reader understand why this sample is representative of the German population. More details on the sampling and the survey procedure should be provided to better inform the reader.
We’re happy that you pointing to this issue. Originally, we decided not to overweight the manuscript with too many details, but now we feel encouraged to do so! Thus, we added an extensive description about our methodological approach and the operational steps of our sampling procedure to achieve representativeness.
Second, I can see the research team devoted considerable efforts to validating data coding. However, it would be a pity not to dig the data a little deeper given this effort. To be specific, the analysis done throughout the paper is basically descriptive in nature, but I think there is much room for more sophisticated statical analyses. For example, who (e.g., people of a certain age, gender, and education level, etc.) are more likely to have the motivators and who are more likely to have deferents of blood donation? You can run some regressions to answer questions like this.
Thank you very much for this appreciative comment. You are right; it would be a pity not to dig the data a little deeper given this effort. However, we have deliberately chosen to present the data in this paper in a descriptive way as we often encountered a need among colleagues for this kind of presentation, which we would like to address here. So first, we wanted to identify motives and barriers in the German population and present it with high resolution. In addition, we are planning 1-2 more papers based on the data, which will provide further in-depth analyses, relations and a stronger link to theory. Unfortunately, putting all information in one paper would be an overload.
Some simple t-tests would also be useful. For example, Table 4 shows the differences in the values of some variables between intenders and non-intenders, but it would be more informative if you show people whether these differences are statistically different across the two types of people. Without a (statistical) benchmark, it's hard to understand whether 5% is a large number or a negligible number in your context.
We thank the reviewer for this recommendation. We added appropriate effect size measures to the main tables of our results section, to get a better grasp of the statistical impact of group-wise differences regarding motives and deterrents.
Finally, while the paper is in general well-written, I found grammatical errors at times. There are also a lot of inconsistencies in citation style. The authors should check the language more carefully to improve the quality of communication.
Thank you for pointing these errors out so we had the chance to edit them as we aim at a well-written manuscript. We have had the article proof read again and improved the citation.

Reviewer 2 Report
The authors conduct a representative survey to elicit quantitative and qualitative information on blood-donation intentions in Germany. The paper is an important contribution to the literature on blood-donations in particular with respect to the qualitatively elicited opinions and intentions. Nevertheless, I have some remarks on how to further strengthen the paper before publication.
- On page 2 the authors claim that their study is needed because other related stories are not up-to-date. It is however not clear why the representative survey by the BZgA in 2018 is not up to date anymore. Related to this, the authors should explain when their survey was conducted.
- The authors claim that their survey was representative for the German population. If this is the case, then a table should compare descriptive statistics to the actual parameters in the population. It is also unclear whether the original sample was representative or resampling was used to achieve representativeness.
- The main results from Table 2 and 4 should be presented in a graph to make it easier to get an overview of the story (the tables can go in an appendix). The authors should also briefly discuss how the overall picture compares to the two related studies previously conducted in Germany.
- Clearly, the biggest contribution of the paper is the systematic analysis of qualitative responses. However, there needs to be some discussion on whether the effort actually lead to more insights compared to related studies that relied more on quantitative scales. The authors should also motivate their method of extracting the categories in more detail. Why for example did they rely on the again subjective judgement of the researcher team and not use one of the widely available classification algorithms for natural language processing?
Minor comments:
- The introduction is framed to highlight that only more blood donors can solve the expected shortage of donations. This is not completely correct, there are other approaches to reduce the shortage such as Patient Blood Management that reduces the demand for blood. The authors should mention this broader picture and highlight why an increase in donations is nevertheless important from a social planner perspective.
- The formatting of subheadings does not seem to work. Either correct formatting needs to be addressed or the subheadings need to be sentences in order to increase readability (e.g. section 2 “Study population and recruitment procedure”).
- Something is wrong with the display of Table 3. The authors should use an actual table and not a screenshot.
Author Response
Dear Reviewer 2, thank you very much for reviewing our manuscript. We appreciated your comments and address them below:
Answer to IJERPH Reviewers – Major Revision
Reviewer II
On page 2 the authors claim that their study is needed because other related stories are not up-to-date. It is however not clear why the representative survey by the BZgA in 2018 is not up to date anymore. Related to this, the authors should explain when their survey was conducted.
Thank you for this comment. We were not aware that our statement could be misunderstood. The BZgA study is of course up-to-date, but limited to a few statements. For this reason, we have added an "or" at the appropriate place in the text, as not all the reasons listed are applicable to both studies.
The authors claim that their survey was representative for the German population. If this is the case, then a table should compare descriptive statistics to the actual parameters in the population. It is also unclear whether the original sample was representative or resampling was used to achieve representativeness.
We’re happy that you pointing to this issue. Originally, we decided not to overweight the manuscript with too many details, but now we feel encouraged to do so! Thus, we added an extensive description about our methodological approach and the operational steps of our sampling procedure to achieve representativeness.
The main results from Table 2 and 4 should be presented in a graph to make it easier to get an overview of the story (the tables can go in an appendix). The authors should also briefly discuss how the overall picture compares to the two related studies previously conducted in Germany.
Presenting the results as a figure is a very good idea, which we have realized with regard to Table 2. For the data in Table 4, on the other hand, we have remained with the presentation as a table. The presentation of a part of the data as a figure would be possible. However, with this format we aim to include all identified motives and barriers in the different groups, which should be a benefit. A comparison of the results to the two existing studies was added in the “Discussion” section.
Clearly, the biggest contribution of the paper is the systematic analysis of qualitative responses. However, there needs to be some discussion on whether the effort actually lead to more insights compared to related studies that relied more on quantitative scales. The authors should also motivate their method of extracting the categories in more detail. Why for example did they rely on the again subjective judgement of the researcher team and not use one of the widely available classification algorithms for natural language processing?
The qualitative approach to assessing motives and barriers is more time-consuming than using quantitative scales. In the context of this explorative study, however, we had the opportunity to take into account all individual motives and barriers and to enable multiple answers. This procedure is particularly recommended if one is interested in motives and barriers of previously neglected groups (e.g. deferred donors), which may differ from the information provided by donors and non-donors. The use of qualitative methods is therefore in line with our aim of improving the mapping of motives and barriers of different donor groups. We added sentences to the strengths and limitation section accordingly.
We are aware that there is already a lot of research in the area of motives and barriers to blood donation. Considering this and to provide a better comparability of our results with previous research, we have used the category system of the meta-analysis as an initial basis for the coding proceudre and differentiated it if necessary. Our coding strategy was not purely explorative, but we took a deductive-inductive approach. For this reason, we did not choose to use classification algorithms for natural language processing.
Minor comments:
The introduction is framed to highlight that only more blood donors can solve the expected shortage of donations. This is not completely correct, there are other approaches to reduce the shortage such as Patient Blood Management that reduces the demand for blood. The authors should mention this broader picture and highlight why an increase in donations is nevertheless important from a social planner perspective.
We thank the reviewer for this important advice. We added a respective sentence in the introduction. You are completely right, there are two major regulation options for saving the blood supply: increasing blood donations, as mentioned in the text, and decreasing blood demand. The blood demand e.g. in the region of Mecklenburg-Western Pomerania (located in rural Northeast Germany) was analyzed in a prospective longitudinal study* since 2005. Here we already observed decreasing transfusion demand. To this point it is unclear, whether there is room for further reductions, especially when the baby-boom-generation is reaching the age groups with high transfusion rates. As further reductions seem to be limited, increasing blood donations remains key. With the help of building donor loyalty, the need for blood donations gets socially plannable, too, which is a huge advantage.
*Schönborn L, Weitmann K, Greinacher A, Hoffmann W. Characteristics of Recipients of Red Blood Cell Concentrates in a German Federal State. Transfus Med Hemother. 2020 Oct;47(5):370-377. doi: 10.1159/000510207. Epub 2020 Sep 22. PMID: 33173455; PMCID: PMC7590768).
The formatting of subheadings does not seem to work. Either correct formatting needs to be addressed or the subheadings need to be sentences in order to increase readability (e.g. section 2 “Study population and recruitment procedure”).
We totally agree the formatting of subheadings does not work well. We were confused by the author guidelines, which led us to this decision first. We now adapted the subheadings.
Something is wrong with the display of Table 3. The authors should use an actual table and not a screenshot.
We updated the Table.

Round 2
Reviewer 1 Report
The authors refused to perform the additional statistical analyses I suggested in the first round of review, which are not at all complicated -- one or two tables would be sufficient. The reason provided by the authors is to "save those for other papers", which I don't think is justifiable.
In fact, the analyses I suggested are a coherent part of this study: Wouldn't the readers wonder "WHO donates blood"? Would they be satisfied by knowing only that "SOMEONE donates blood"? What if you are the reader of such a paper? I, as a reader, would not be satisfied.
Author Response
Dear Reviewer, thank you for pointing this out again. We have added two suitable tables (Table 2a & 2b) and phrases in the results section and the discussion. We hope that we have found a good compromise.
